# An Immunohistochemical Study of MAGE Proteins in Hepatocellular Carcinoma

**DOI:** 10.3390/diagnostics14151692

**Published:** 2024-08-05

**Authors:** Stylianos Tologkos, Vasiliki Papadatou, Achilleas G. Mitrakas, Olga Pagonopoulou, Grigorios Tripsianis, Triantafyllos Alexiadis, Christina-Angelika Alexiadi, Antonios-Periklis Panagiotopoulos, Christina Nikolaidou, Maria Lambropoulou

**Affiliations:** 1Laboratory of Histology-Embryology, Medical School, Democritus University of Thrace, 68100 Alexandroupolis, Greece; steltolo@gmail.com (S.T.); vasipapadatou@gmail.com (V.P.); alexiadistr@gmail.com (T.A.); alexiadichr@gmail.com (C.-A.A.); axilleas_mt@hotmail.com (A.-P.P.); mlambro@med.duth.gr (M.L.); 2Laboratory of Neurophysiology, Medical School, Democritus University of Thrace, 68132 Alexandroupolis, Greece; opagonop@med.duth.gr; 3Laboratory of Medical Statistics, Medical School, Democritus University of Thrace, 68132 Alexandroupolis, Greece; gtryps@med.duth.gr; 4Laboratory of Pathology, Ippokrateio General Hospital of Thessaloniki, 54642 Thessaloniki, Greece; christinapathologist@gmail.com

**Keywords:** hepatocellular carcinoma, HCC, MAGE family, MAGE-C1, MAGE-C2

## Abstract

Hepatocellular carcinoma (HCC) is one the most common primary malignancies with high mortality and morbidity. The melanoma-associated antigen (MAGE) gene family includes several genes that are highly expressed in numerous human cancers, making many of them part of the cancer-testis antigen (CTA) family. MAGE-C1 is expressed in various malignancies but is absent in normal cells, except for the male germ line. Its presence is associated with a worse prognosis, increased tumor aggressiveness, and lymph node invasion. Similarly, MAGE-C2 is linked to the development of various malignant tumors. Despite these associations, the roles and mechanisms of MAGE-C1/MAGE-C2 in HCC remain unclear. This study aimed to evaluate the expression of MAGE-C1 and MAGE-C2 in HCC and correlate it with clinicohistological characteristics. Our findings indicated that MAGE-C1 expression is associated with a higher number of nodules, elevated AFP levels, HBV or HCV positivity, older age, male sex, and lymph node invasion. MAGE-C2 expression was correlated with these characteristics and the presence of cirrhosis. These results align with the limited literature, which suggests a correlation between MAGE expression and older age and HBV infection. Consequently, our study suggests that MAGE-C1 and MAGE-C2 are promising novel biomarkers for prognosis and potential therapeutic targets in HCC.

## 1. Introduction

Hepatocellular carcinoma (HCC) is the most common primary liver malignancy, constituting approximately 90% of all cases of liver malignancy. Primary liver cancer has high mortality rates (8%), making HCC a significant cause of death, with more than half a million deaths each year being attributed to it. Moreover, HCC is the third-leading cause of death due to cancer. Thus, the high mortality rate and the lack of an effective treatment plan make hepatocellular carcinoma an enormous challenge for the medical community [1,2,3,4]. Early-stage symptoms of HCC are not specific, making it hard to diagnose at an early stage in order to perform surgical resection, and thus, surgical resection can be performed in only one-third of the patients. Moreover, even though various therapeutic protocols, such as radiofrequency ablation, liver transplantation, and molecular therapy, have been developed to treat HCC, still, the overall survival is considerably poor [5,6]. Currently, the most widely used biomarker for early detection is AFP; however, a large number of AFP-negative HCC patients exist (60% sensitivity) [7], making the need for more efficient novel prognostic biomarkers imminent, considering HCC is mostly lethal because it is extremely difficult to diagnose in early stages. Even in patients who have undergone curative resection, the 5-year recurrence rate is relatively high (approximately 70%) [8,9]. This clearly indicates the need to better understand the molecular background of these tumors so we can develop better therapeutic targets.

The melanoma-associated antigen (MAGE) gene family contains various genes that are highly expressed in many human malignancies, and thus, many are also members of the cancer-testis antigen (CTA) family. As a result, they have gained interest as both biomarkers and therapeutic targets [10,11]. Homology studies have discovered approximately 60 members in the family, which are found conserved in all eukaryotic organisms and also share a common domain of about 200 amino acids [12,13]. MAGE genes are categorized into two types, type I, which consists of subfamilies A–C, and type II, which consists of subfamilies D–H and L. Type I MAGE genes are clustered on the X chromosome and are those that are considered CTAs, while type II genes are not clustered on the X chromosome and are not tissue specific [14,15,16]. Until this day, the expression of various MAGE genes has been correlated with different cancers, including prostate cancer [17,18], gliomas [19,20], colon cancer [21,22], lung cancer, and, particularly, non-small-cell lung cancer [23,24,25], as well as breast cancer [26,27,28]. MAGE members are considered not only potential biomarkers but also immunotherapeutic targets in various types of cancers [29]. Brisar et al. showed that the intraperitoneal administration of MAGE-A1 peptides on engineered non-lytic M13 bacteriophages can induce specific anti-MAGE antibodies and CTL cytotoxicity, highlighting the potential of filamentous bacteriophages in cancer immunotherapy [30,31,32]. It is noteworthy that MAGE members are also associated with other pathways, which may be significant for improving therapeutic outcomes [33].

Taking all into consideration, the association between MAGE family members and cancer prognosis and treatment is undeniable [34,35,36,37,38]. Thus, this study was designed to detect MAGE-C1 and MAGE-C2 expression in hepatocellular carcinoma tissues from a large number of patients. The expression levels of these two MAGEs were correlated with various clinical and histological parameters. A positive correlation between the expression levels of MAGE-C1 and MAGE-C2 and clinicopathological parameters, as well as specific characteristics of the disease, could indicate that MAGEs serve as biomarkers.

## 2. Materials and Methods

### 2.1. Tumor Samples

We used 57 samples from patients with hepatocellular carcinoma, in collaboration with the Hippokration General Hospital of Thessaloniki. Tissue samples were surgically removed during the period from 2013 to 2022 and immediately placed in 10% neutral-buffered formalin for fixation at room temperature for 24 h. Following fixation, the tissues were dehydrated through a graded series of ethanol, cleared in xylene, and then embedded in paraffin wax. The paraffin-embedded tissues were then allowed to solidify, creating FFPE blocks suitable for further histological analysis. All samples were transferred to the Laboratory of Histology-Embryology at the Medical School of Democritus University of Thrace in Alexandroupolis, Greece, for sectioning, immunostaining, evaluation, and analysis of results, while maintaining proper conditions to ensure sample integrity. The samples were selected to include a diverse patient cohort with a mix of gender and a history of HBV or HCV infection.

Of the 57 patients, 40 were male and 17 were female, with 12 patients having a history of HBV or HCV infection. The detailed characteristics of the patient cohort, including age, tumor stage, and other relevant clinical information, are summarized in Table 1.

### 2.2. Ethical Considerations

The study was conducted in accordance with the ethical standards of the Declaration of Helsinki and its later amendments. All patients provided informed consent prior to the collection of samples, and the study protocol was approved by the Institutional Review Board.

### 2.3. Immunohistochemical Staining

Serial 3 μm sections of tissue blocks were obtained using a Leica RM2030 automated microtome (Leica Microsystems, Wetzlar, Germany). All tissue samples were stained using the peroxidase method (EnVision FLEX, Mouse/Rabbit Detection System, High pH; DAKO, Carpinteria, CA, USA). Antigen retrieval was achieved using EnVision FLEX Target Retrieval Solution High pH (50×) (catalog no. K8004). Slides were incubated with 1:200-diluted primary rabbit anti-MAGE C1 and C2 antibodies (Abcam, ab137524, Cambridge, UK) at 4 °C overnight. In parallel, control slides were incubated with non-immunized rabbit serum (negative control), while a positive control from the human male germ line was always used. Visualization of the antibody–antigen complex was performed with the EnVisionTM FLEX diaminobenzidine (DAB) chromogen (DM827, DAKO, Carpinteria, CA, USA). Sections were counterstained with Mayer’s hematoxylin, mounted, and examined using a Nikon Eclipse 50× microscope (Nikon Instech Co., Ltd., Kawasaki, Japan). The staining results were evaluated based on the percentage of staining in cells. MAGE-C1/MAGE-C2 immunoreactivity was scored for cytoplasmic or nuclear expression. Sections were graded as negative (0) when 10% of the cells were stained, low (1) when 10–30% of the cells were stained, moderate (2) when 30–70% of the cells were stained, and high (3) when 70% of the cells were stained.

### 2.4. Statistical Analysis

Statistical analysis of the data was performed using IBM Statistical Package for Social Sciences (SPSS) version 19.0 (IBM Corp., Armonk, NY, USA). The chi-square test and the Mann–Whitney U test were used to evaluate any potential association of MAGE-C1 and MAGE-C2 expression, as well as their co-expression, with patients’ clinicopathological parameters.

Odds ratios (ORs) and their 95% confidence intervals (CIs) were estimated as a measure of the association of MAGE-C1 and MAGE-C2 expression with the patients’ parameters using logistic regression models. As an indicator of survival, disease-specific survival (DSS; time from cancer diagnosis to death from cancer) was investigated. Survival rates were calculated with the Kaplan–Meier method, and the statistical difference between survival curves was determined with the log-rank test. Multivariate Cox proportional hazards regression analysis, using a backward selection approach, was performed to explore the independent effect of MAGE-C1 and MAGE-C2 expression, as well as their co-expression, and other clinicopathological parameters on survival indicators. All tests were two tailed, and statistical significance was considered for *p* values < 0.05. The table containing the complete characteristics of the statistical analysis is presented in the Appendix A.

## 3. Results

### 3.1. Association of MAGE Expression Levels and Different Clinicopathological Parameters

In this study, we assessed the expression levels of MAGE-C1 and MAGE-C2 through immunohistochemical staining. Images from these experiments were captured following the experimental protocol outlined previously.

The IHC pattern of MAGE-C1 expression is exhibited in Figure 1.

MAGE-C1 showed cytoplasmic expression. The levels of MAGE-C1 were found to be higher in female patients compared to male patients. Additionally, patients over the age of 70 and those with a history of HBV or HCV infection presented higher levels of MAGE-C1. The presence of lymph node invasion, high AFP levels, and an increased number of nodules were statistically significantly associated with a higher expression of MAGE-C1. Analytically, patients with G3 tumors constituted the vast majority of those (91.3%) who showed high expression levels of MAGE-C1. Moreover, 87.2% of the samples positive for MAGE-C1 expression presented lymph node invasion, and there was a positive correlation with the number of nodules (Figure 2).

MAGE-C2 exhibited the same localization and pattern as MAGE-C1 in female compared to male patients and in patients with a history of HBV or HCV infection. The IHC pattern of MAGE-C2 expression is exhibited in Figure 3. However, the association with age showed a controversial pattern for MAGE-C2 compared to MAGE-C1, as MAGE-C2 was expressed in higher levels in patients in the age group of <70. Tumors classified as higher grade exhibited elevated expression levels of MAGE-C2. In addition, 78.3% of the samples positive for MAGE-C2 expression were classified as G3. Furthermore, 59% of positive samples presented lymph node invasion and an increased number of nodules (Figure 4). In addition to the aforementioned results, higher expression levels of MAGE-C2 were linked to the presence of cirrhosis.

### 3.2. Association of MAGE Expression Levels and Overall Survival Rates of Patients

The 6-month survival rate for all patients with HCC was 57%, and the 12-month survival rate was 24%, with a mean survival time of approximately 8 months (Figure 5A). The group of patients with positive expression of MAGE-C1 showed a reduction in the 6-month survival rate to approximately 45% and in the 12-month survival rate to around 11%. In contrast, patients with negative expression of MAGE-C1 showed survival rates of 92.86% and 54.17% at 6 and 12 months, respectively. Moreover, the mean survival time for the group with positive MAGE-C1 expression was 6.56 months compared to 12.42 months for the group with negative expression. The overall survival rate of patients with hepatocellular carcinoma decreased significantly with higher levels of MAGEs (Figure 5B).

The same pattern was observed for overall survival rates for the group of patients with positive expression of MAGE-C2. Analytically, the 6-month survival rate reduced to 36% and the 12-month survival rate to around 6.75% compared to the group that showed negative expression of MAGE-C2. The mean survival time for the group with positive MAGE-C2 expression was 5.38 months compared to 10.31 months for the group with negative expression (Figure 5C). When MAGE-C1 and MAGE-C2 were expressed simultaneously, it led to a larger decrease in the overall survival rate (Figure 5D).

## 4. Discussion

MAGE-C1 is correlated with various malignancies, such as melanoma, as we prostate cancer, breast cancer, and thyroid cancer, but it is absent in normal cells except for the male germ line [27,28,29]. Its expression has been associated with a worse prognosis, higher tumor aggressiveness, and lymph node invasion in various malignancies. Similarly, MAGE-C2, which shares significant homology with MAGE-C1, has been associated with the development of various malignant tumors. However, the actual role and mechanism of MAGE-C1 and MAGE-C2 proteins in hepatocellular carcinoma (HCC) remain unclear. Given the limited literature on this subject, our study aimed to evaluate the expressions of MAGE-C1 and MAGE-C2 in HCC and correlate these expressions with clinical and histological characteristics.

Our results indicated that MAGE-C1 expression is higher in female patients compared to male patients, and it is elevated in patients over the age of 70 and those with a history of HBV or HCV infection. The expression of MAGE-C1 was significantly associated with the presence of lymph node invasion, higher AFP levels, and an increased number of nodules. Notably, patients with high-grade tumors constituted the vast majority (91.3%) of those showing high expression levels of MAGE-C1. Furthermore, 87.2% of the samples positive for MAGE-C1 expression presented lymph node invasion, and there was a positive correlation with the number of nodules. These findings are consistent with the limited existing literature, which also shows a correlation between higher MAGE-C1 expression and the older age of patients [33].

Similarly, MAGE-C2 expression exhibited a comparable pattern to MAGE-C1 in terms of higher expression in female patients and those with a history of HBV or HCV infection. However, unlike MAGE-C1, MAGE-C2 was expressed at higher levels in patients younger than 70. Tumors classified as higher grade also exhibited elevated expression levels of MAGE-C2, with 78.3% of positive samples falling into this category. Additionally, 59% of positive samples presented lymph node invasion and an increased number of nodules. Importantly, higher MAGE-C2 expression was linked to the presence of cirrhosis, an association not observed with MAGE-C1.

Survival analysis revealed that patients with positive MAGE-C1 expression have significantly lower 6-month and 12-month survival rates (45% and 11%, respectively) compared to those with negative expression (92.86% and 54.17%, respectively). The mean survival time for the group with positive MAGE-C1 expression was 6.56 months compared to 12.42 months for the group with negative expression. This trend was also observed for MAGE-C2, with positive expression linked to reduced survival rates at 6 months (36%) and 12 months (6.75%) and a mean survival time of 5.38 months compared to 10.31 months for those with negative expression. Notably, simultaneous expression of MAGE-C1 and MAGE-C2 led to a more significant decrease in overall survival rates.

Our findings are supported by previous studies. Sideras et al. demonstrated a positive relationship between MAGE protein expression and HBV infection [34]. This aligns with our observations of a significant association between MAGE expression and HBV/HCV positivity.

We can conclude that our study provides compelling evidence that MAGE-C1 and MAGE-C2 are not only novel biomarkers for prognosis but also potential therapeutic targets in HCC. The significant correlations between MAGE expression and various clinical and histological parameters, including survival outcomes, underscore their potential utility in clinical practice. It is highly significant to study the correlation between alterations in MAGE expression levels and the response of cancer cells to various therapies. Additionally, the association of MAGE proteins with the immune system response may be a key factor for developing immunotherapies against cancers that express these protein families. Further research is needed to elucidate the underlying mechanisms by which MAGE-C1 and MAGE-C2 contribute to HCC progression and to explore their roles in targeted therapies.

## Figures and Tables

**Figure 1 diagnostics-14-01692-f001:**
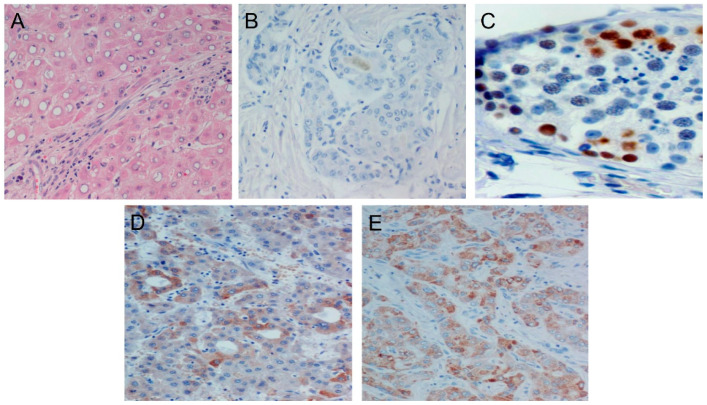
Distinct staining patterns of the control and samples. (**A**) Hepatocellular carcinoma H&E staining, (**B**) HCC MAGE-C1 negative control, (**C**) MAGE-C1 positive control (testis), (**D**) MAGE-C1 moderate expression—IHC staining, and (**E**) MAGE-C1 high expression—IHC staining.

**Figure 2 diagnostics-14-01692-f002:**
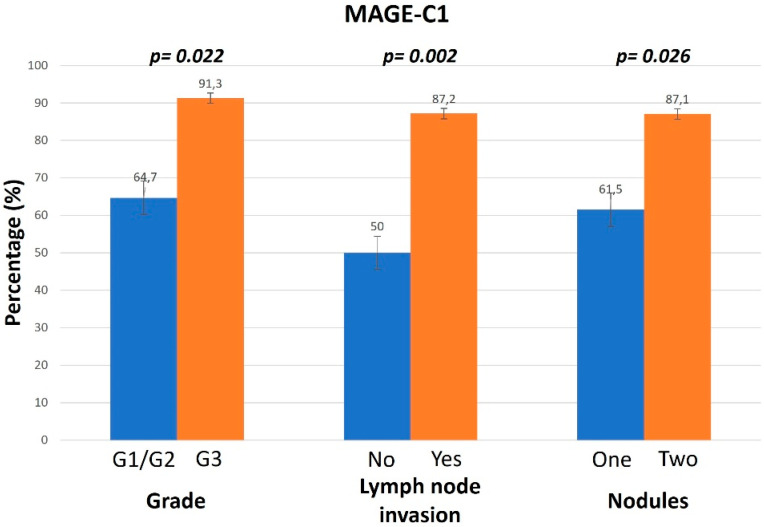
Positive (>30%) MAGE-C1 expression in relation to grade, lymph node invasion, and nodules.

**Figure 3 diagnostics-14-01692-f003:**
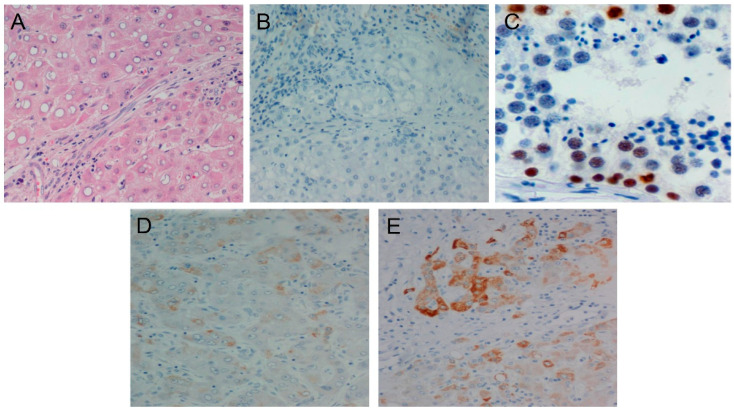
Distinct staining patterns of the control and samples. (**A**) Hepatocellular carcinoma H&E staining, (**B**) HCC MAGE-C2 negative control, (**C**) MAGE-C2 positive control (testis), (**D**) MAGE-C2 low expression—IHC staining, and (**E**) MAGE-C2 moderate expression—IHC staining.

**Figure 4 diagnostics-14-01692-f004:**
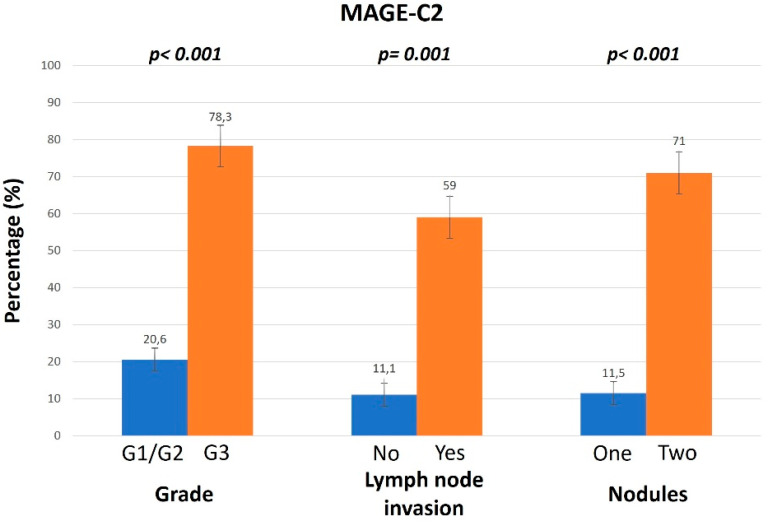
Positive (>30%) MAGE-C2 expression in relation to grade, lymph node invasion, and nodules.

**Figure 5 diagnostics-14-01692-f005:**
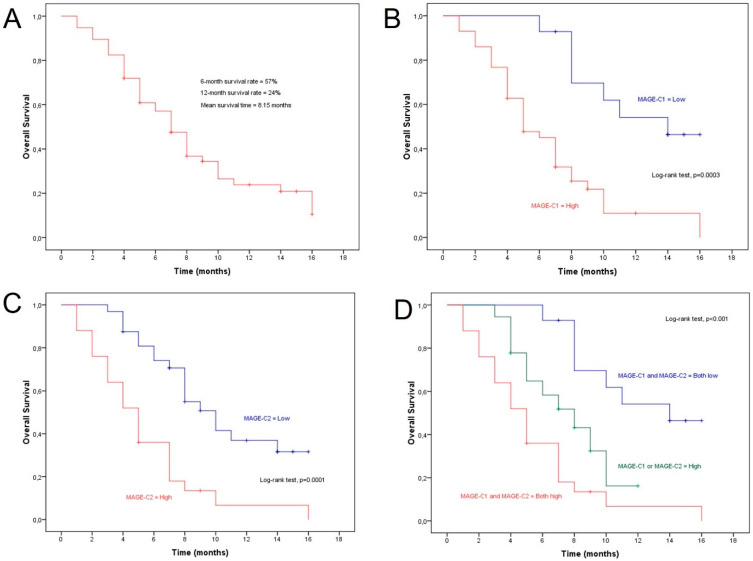
Association of MAGE expression levels and overall survival rates of patients. (**A**) Overall survival of patients with HCC. (**B**) Overall survival of patients with HCC in relation to MAGE-C1 expression. (**C**) Overall survival of patients with HCC in relation to MAGE-C2 expression. (**D**) Overall survival of patients with HCC in relation to MAGE-C1/MAGE-C2 co-expression.

**Table 1 diagnostics-14-01692-t001:** Clinical and histopathological characteristics of patients with hepatocellular carcinoma.

	Total Sample
**Sex**	
Female	17 (29.8)
Male	40 (70.2)
**Age**	
<70 years	23 (40.4)
≥70 years	34 (59.6)
**HBV or HCV**	
Negative	45 (78.9)
Positive	12 (21.1)
**Grade**	
G1/G2	34 (59.6)
G3	23 (40.4)
**Lymph Node Invasion**	
No	18 (31.6)
Yes	39 (68.4)
**Νodules**	
1	26 (45.6)
2	31 (54.5)
**AFP**	
Low	25 (43.9)
High	32 (56.1)
**Cirrhosis**	
No	43 (75.4)
Yes	14 (24.6)
**Surgery**	
No	12 (21.1)
Yes	45 (78.9)
**Status**	
Alive	16 (28.1)
Dead	41 (71.9)

## Data Availability

The original contributions presented in the study are included in the article/Appendix A, further inquiries can be directed to the corresponding author.

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
