# Peer review of "An Immunohistochemical Study of MAGE Proteins in Hepatocellular Carcinoma"

_diagnostics, 2024, doi:10.3390/diagnostics14151692_

Round 1

Reviewer 1 Report

Comments and Suggestions for Authors

The article "An immunohistochemical study of MAGE proteins in hepatocellular carcinoma" focuses on the evaluation of the expression of MAGE-C1 and MAGE-C2 in HCC and correlation of the expression with clinico-histologic features.Undoubtedly, the study is of scientific interest. MAGE is currently poorly investigated, despite the fact that correlations between poor prognosis and expression of some members of this tumor-associated antigen family have been described in the literature, including for HCC. By selecting MAGE-C1 and MAGE-C2 as study representatives, the authors contribute to the MAGE family study. Nevertheless, there are a number of comments to the publication: 

1. The article is very carelessly written. In line 77, there is no period at the end of the sentence. There is an extra period in line 93. Figures 2 and 4 are poorly designed: the inscription MAGE-C1 and MAGE-C2 are oddly placed, the figures themselves are of poor quality, one of them has a cursor. In some places in the text, the word lymph is capitalized unnecessarily. In lines 146-148 the text is italicized unnecessarily. Line 162 does not have a number at the header.

2. The study itself does not show whether the localization of the proteins in the cells was investigated. The expression of proteins was also not confirmed by PCR.

3. The article states that MAGE family proteins are a promising target for therapy, but the article does not give any reason for this: the localization as well as the function of the proteins are not known. It would be more reasonable to state that tumor typing by MAGE expression is promising for prognostic purposes and for possible correction of therapy.

Thus, it is reasonable to send the article for major revision.

Author Response

Firstly, we would like to thank the reviewers for their valuable comments and interesting recommendations.

REVIEWER 1

The article "An immunohistochemical study of MAGE proteins in hepatocellular carcinoma" focuses on the evaluation of the expression of MAGE-C1 and MAGE-C2 in HCC and correlation of the expression with clinico-histologic features.Undoubtedly, the study is of scientific interest. MAGE is currently poorly investigated, despite the fact that correlations between poor prognosis and expression of some members of this tumor-associated antigen family have been described in the literature, including for HCC. By selecting MAGE-C1 and MAGE-C2 as study representatives, the authors contribute to the MAGE family study. Nevertheless, there are a number of comments to the publication:

  1. The article is very carelessly written. In line 77, there is no period at the end of the sentence. There is an extra period in line 93. Figures 2 and 4 are poorly designed: the inscription MAGE-C1 and MAGE-C2 are oddly placed, the figures themselves are of poor quality, one of them has a cursor. In some places in the text, the word lymph is capitalized unnecessarily. In lines 146-148 the text is italicized unnecessarily. Line 162 does not have a number at the header.

Our response: Thank you for your suggestions. We have already addressed all the points you referred to in the text. Additionally, the images have been improved for better clarity and design.

  1. The study itself does not show whether the localization of the proteins in the cells was investigated. The expression of proteins was also not confirmed by PCR.

Our response: Thank you for your comment. MAGEs showed cytoplasmic expression, and we have added this information to the text. This paper presents preliminary results based on an immunohistochemical study. We have planned an additional project on MAGEs expression at the gene and protein levels for the future.

  1. The article states that MAGE family proteins are a promising target for therapy, but the article does not give any reason for this: the localization as well as the function of the proteins are not known. It would be more reasonable to state that tumor typing by MAGE expression is promising for prognostic purposes and for possible correction of therapy.

Our response: Thank you for your feedback. I understand your concern regarding the justification for targeting MAGE family proteins in therapy. While the article may not explicitly detail the about the function of these proteins, the rationale for considering MAGE family proteins as therapeutic targets often stems from their cancer/testis antigen characteristics, which are typically restricted to tumors and absent in normal tissues. This selective expression makes them attractive biomarkers or candidates for targeted therapy, minimizing damage to normal cells.

However, I agree that the current evidence more robustly supports the use of MAGE expression for tumor typing and prognostic purposes. This approach can indeed aid in the stratification of patients and the customization of therapeutic strategies. We have revised the statement to reflect this perspective more accurately. Moreover, in our next steps, we will try to alter the expression levels of MAGEs to test the sensitization of cancer cells to various therapies. Thank you for pointing this out.

Reviewer 2 Report

Comments and Suggestions for Authors

This study was well designed and well written with surprising results. There were some comment for improving manuscript:

1- The study setting should be added in the method section.

2- The date of study and patients enrolment must be added in the method section.

3- The sampling method must be mentioned in the method section.

4- As an important comment, the treatment regime plays an important role in survival and must be added and analyzed, and then related results and discussion must be added in manuscript. For generalizability of results, this item must be considered, except clinicopathologic items.

Author Response

This study was well designed and well written with surprising results. There were some comment for improving manuscript:

1- The study setting should be added in the method section.

Our response: Thank you for your suggestion. It has been added in the paragraph 2.1 Tumor samples.

2- The date of study and patients enrolment must be added in the method section.

Our response: Thank you for your suggestion. It has been added in the paragraph 2.1 Tumor samples.

3- The sampling method must be mentioned in the method section.

Our response: Thank you for your suggestion. It has been added in the paragraph 2.1 Tumor samples.

4- As an important comment, the treatment regime plays an important role in survival and must be added and analyzed, and then related results and discussion must be added in manuscript. For generalizability of results, this item must be considered, except clinicopathologic items.

Our response: Thank you for your insightful comment. We agree that the treatment regimen is a crucial factor influencing patient survival and should be considered in our analysis. As this manuscript presents preliminary results, we have already submitted a project proposal to extend our analyses to include treatment regimens. We plan to incorporate a detailed examination of how these regimens impact survival in relation to MAGE expression alterations. This future work will enhance the generalizability of our findings and provide a more comprehensive understanding of the relationship between MAGE proteins and patient outcomes. We believe that this additional research will contribute significantly to improving therapeutic strategies for patients with hepatocellular carcinoma (HCC).

Round 2

Reviewer 1 Report

Comments and Suggestions for Authors

The authors have considered the comments and made the necessary changes to the article. I recommend it for publication in its present form after text correction.